# 3D-Printed Soft Structure of Polyurethane and Magnetorheological Fluid: A Proof-of-Concept Investigation of its Stiffness Tunability

**DOI:** 10.3390/mi10100655

**Published:** 2019-09-29

**Authors:** Seong-Woo Hong, Ji-Young Yoon, Seong-Hwan Kim, Sun-Kon Lee, Yong-Rae Kim, Yu-Jin Park, Gi-Woo Kim, Seung-Bok Choi

**Affiliations:** Smart Structure and Systems Laboratory, Department of Mechanical Engineering, Inha University, Incheon 22212, Korea; ghdtjddn3@gmail.com (S.-W.H.); ji_young62@naver.com (J.-Y.Y.); neumann9177@naver.com (S.-H.K.); sun@inha.ac.kr (S.-K.L.); rae9609@naver.com (Y.-R.K.); pyjn5059@naver.com (Y.-J.P.); gwkim@inha.ac.kr (G.-W.K.)

**Keywords:** three-dimensional printing, magnetorheological fluid, magnetic responsivity, soft structure, tunable elastic stiffness

## Abstract

In this study, a soft structure with its stiffness tunable by an external field is proposed. The proposed soft beam structure consists of a skin structure with channels filled with a magnetorheological fluid (MRF). Two specimens of the soft structure are fabricated by three-dimensional printing and fused deposition modeling. In the fabrication, a nozzle is used to obtain channels in the skin of the thermoplastic polyurethane, while another nozzle is used to fill MRF in the channels. The specimens are tested by using a universal tensile machine to evaluate the relationships between the load and deflection under two different conditions, without and with permanent magnets. It is empirically shown that the stiffness of the proposed soft structure can be altered by activating the magnetic field.

## 1. Introduction

The development of flexible or soft structures based on smart materials has attracted considerable interest owing to their effectiveness for vibration control through the tunable viscoelastic properties by external stimuli such as magnetic fields. Electrorheological fluids (ERFs), magnetorheological fluids (MRFs), and magnetorheological elastomers (MREs) are some of the smart materials which are frequently used for the vibration control of flexible structures. The ERF is a special suspension consisting of fine non-conducting particles (5–50 μm) and a carrier liquid which is electrically insulated. The apparent viscosity of the ERF can be adaptively tuned by applying an electric field; its tuning capability is reversible. The response to a viscosity change is very fast (3–5 ms). Therefore, ERFs are frequently used as core components of sandwich structures (beam, shell, plate) to achieve excellent controllability of modal characteristics by applying different electric fields to the fluid domain. Therefore, an effective vibration control is provided by switching the natural frequency based on the excitation frequency to avoid a resonance behavior [1,2,3,4]. Similar to the ERF, the MRF is a smart material whose rheological characteristics can be tuned by an external magnetic field. The MRF consists of iron particles and a base oil such as silicone oil. The iron particles are randomly distributed without a magnetic field and can form chain-like structures upon the application of a magnetic field. As the force of the chain-like structure is larger than that of the ERF, vibration control by using MRFs in soft or flexible structures has been extensively investigated [5,6,7,8]. Besides the ERFs and MRFs, MREs can be also used in the fabrication of flexible structures with superior vibration control capabilities. Unlike the ERF and MRF, the MRE is a solid whose stiffness and damping properties can be controlled by applying an external magnetic field to the iron particles distributed in the matrix, such as in silicone rubber. Therefore, extensive studies on flexible structures incorporated with MRE layers have been carried out for vibration control [9,10,11,12].

Most studies on soft or flexible structures have focused on vibration control or/and vibration suppression by using the inherent field-dependent characteristics of smart ERFs, MRFs, and MREs. On the contrary, no extensive studies have been carried out on soft or flexible structures for stiffness tuning or control. Wagner and Bauer [13] have summarized various soft and flexible materials that can be integrated with electronic circuits. They have reported that stretchable electronic surfaces can combine viscoelastic, plastic, and brittle materials with large differences (several orders of magnitude) in Young’s modulus, which are directly related to the stiffness of the soft or flexible structures. Recently, Qi et al. [14] have developed an advanced MRE with a higher storage modulus than that of the conventional MRE at the same magnetic field intensity. They have experimentally demonstrated the enhanced dynamic viscoelastic properties of the proposed MRE, which is a key factor in achieving a wide range of stiffness tuning in the structures. Shan et al. [15] have proposed a tunable composite by using a polydimethylsiloxane elastomer with a low-melting-point metal solder and have evaluated the stress–strain behaviors at both nonactivated and activated states. The Young’s modulus was tuned by four orders of magnitude when the composite was electronically activated. In other words, the stiffness of the soft composite could be easily tuned by applying different aspects of the external field, such as the magnetic field, temperature, and electric field. Recently, Allen et al. [16] have proposed a smart composite for soft robots whereby a localized geometric patterning of smart materials could provide discrete levels of stiffness through the combinations of smart materials. They have fabricated a shape-memory-alloy-(Nitinol)-based composite finger and demonstrated the stiffness controllability by applying different temperatures as an external input source.

In this study, a flexible soft structure, whose stiffness can be tuned, is proposed, and its potential is validated by a test-bed experiment. The proposed soft structure is fabricated by using an MRF embedded in channels in thermoplastic polyurethane (TPU). Beam types of specimens with 12 channels and three channels, respectively, are fabricated by three-dimensional (3D) printing for both channel patterning and MRF injection with two nozzles. In the fabrication, a filament wire is used to provide sealing to avoid MRF leakage. A tensile test is then carried out to evaluate the relationship between the applied load and corresponding deflection. In addition, the stress–strain curve is measured without and under a magnetic field to evaluate the tuning range of the Young’s modulus of the proposed soft structure. The presented preliminary results demonstrate the stiffness tunability of the proposed soft structure, which can be further analyzed for various applications including the discrete-channel-control-based bio-haptic patch. 

## 2. Fabrication of the Beam Structure

Two types of specimen are fabricated, and their configurations and geometric dimensions are shown in Figure 1. Their total lengths and widths are 60 and 30 mm, respectively. To fabricate channels by 3D printing, the thickness of the TPU is set to 12 mm. The number and depth of the channels are determined by considering the flexible rigidity of the structure and yield stress of the used MRF (132 DG, Lord Company, Cary, NC, USA). The analysis is performed by using the finite element method by investigating the stresses of the structure under and without the magnetic field. Figure 2 shows scanning electron microscopy (SEM, Horiba EX-250, Kyoto, Japan) images of the MRF in the absence or presence of the magnetic field. The particles of the MRF are randomly distributed in the absence of the magnetic field (Figure 2a). In this case, the MRF can be treated as a Newtonian fluid with a constant viscous coefficient. However, upon the application of the magnetic field, the particles rapidly form a chain-like structure (Figure 2b) and the phase of the MRF changes from liquid to solid-like with the field-dependent apparent viscosity. The phase change is reversible. Its response time is in the order of milliseconds. Therefore, the stiffness tuning of the proposed structure can be realized in a real-time manner. Notably, the stiffness tuning of the whole structure or each channel can be achieved with an appropriate magnetic core circuit. However, as a preliminary analysis, the stiffness change of the whole structure is considered in this study. The salient characteristic of the fast response of the MRF can provide several benefits in practical applications in which high-frequency components should be controlled. Notably, a thinner TPU (skin structure) provides a softer structure with a wider stiffness tuning range. The fused deposition modeling (FDM) technique is used to obtain square channels in the TPU.

Figure 3 shows the schematics of the configuration (a) and fabrication by 3D printing (b). The specimen is deposited by the FDM process according to the layer-by-layer deposition method. The nozzle and syringe-type injector are moved independently and discharge TPU and MRF, respectively, so that the two materials coexist in the specimen. The process parameters include a printing temperature of 210 °C, a temperature of the heated bed of 80 °C, a printing speed of 30 mm/s, a nozzle diameter of 0.4 mm, a syringe diameter of 0.2 mm, and a layer height of 0.2 mm.

As the first step, the outside wall is stacked by the polymer, followed by the filling of MRF. In this process, the deflection of the upper side of the wall is prevented by adjusting both the surface tension and movement speed of the nozzle. In addition, the sealing issue (to avoid MRF leakage) is overcome by using the thermal fusion method, in which the stacking sequence is completed with the adhesive on the TPU. The total volume to be filled in the 12 channels is calculated, and then the same amount of MRF is charged to avoid the cavity problem. Figure 4 shows a photograph of the 3D printing equipment illustrated in Figure 3. The TPU is machined with the channels by using a nozzle, while the MRF is filled by using another nozzle. Figure 5 shows a schematic and a photograph of the fabricated structure based on the smart MRF filled in the channels of the TPU. It is noted that the first specimen (a) is made to have 12 channels along the width-wise direction, and the second specimen (c) is made to have channels along the length-wise direction. The reason why a different channel configuration is chosen is to investigate the stiffness change in terms of the channel direction, which creates a different magnetic field as a result of the MRF filled in the channels. In the subsequent section, the stiffness change of this structure is investigated by applying a magnetic field to the MRF domains of the channels.

## 3. Experiment and Results

Prior to the test, different approaches of magnetic field generation are considered, such as the use of a circuit core and use of a permanent magnet (PM). The former method is expected to be more effective to achieve a wider stiffness tuning range, but its practical realization is complex. Thus, a PM is used in this study to generate a magnetic field around the MRF domain. Figure 6b–d shows the schematics of the magnetic field distributions of the specimens, which depend on the locations of PMs (top/bottom, lateral). Neodymium-type PMs are used in this study at the top/bottom and lateral locations, which provide fields of 0.2 and 0.3 T, respectively. The diameter and thickness of the top/bottom PM are 10 and 3 mm, while those of the side PM are 10 and 4 mm, respectively. The specimen is fixed by using the fixture shown in Figure 6a. A universal tensile machine, providing a maximum loading force of 200 kgf, is used to obtain the relationship between the force and deflection. The universal tensile machine contains an accurate load cell for force measurement, which has high linearity and repeatability (below 0.03%). For the deflection measurement, an incremental encoder is embedded in the servo motor whose measurement variation is below 0.01%. For data collection, the test is repeated three times and the average value is presented. In the repeated test, the measurement error is very low. Considering the stiffness of the fabricated structure, the loading force is varied in the range of 10–100 kgf. The maximum tensile stroke and testing speed are set to 2 mm and 1 mm/min, respectively, to avoid structure breakage.

Figure 7 presents the stiffness change in the proposed beam structure with PMs attached at the top and bottom (Figure 6). The stiffness is not linear regardless of the magnetic field. When PMs are attached on the top and bottom, the average increment ratios of the stiffness from the origin to strokes of 1 and 2 mm are approximately 22.9% and 13.9% upon the application of the magnetic field, while those for the configuration with PMs attached on the sides are approximately 24.7% and 20.1% (see Figure 8), respectively. When PMs are attached to the top and bottom, the stiffness is constant (approximately 20 N/mm) regardless of the magnetic field application in the tensile length range of 1 to 2 mm. When the magnets are attached to the sides, the increase in the magnetic field is constant (23 N/mm). The maximum ratio of the stiffness increment is higher than that in the case in which PMs are attached at the top and bottom. This indicates that stronger PMs being attached at the sides generates a higher intensity of the magnetic field than that obtained by PMs at the top and bottom (Figure 6). Figure 9 presents the relationship between the stress and strain corresponding to Figure 8. It can be seen from the curves that the stress of 110.74 kPa without activating PM is increased up to 135.82 kPa by activating PM. Thus, the stress is increased up to 16.3% by applying the magnetic field using PM. This directly indicates that the Young’s modulus of the proposed flexible beam can be tuned by the intensity of the magnetic field. Figure 10 presents the stiffness change of specimen 2 with PMs attached at the top and bottom (Figure 6d). The average stiffness increment ratios from the origin to the strokes of 1 and 2 mm are approximately 39.78% and 23.10% upon the application of the magnetic field, respectively. Specimen 2 exhibits considerably larger increases in both sections than those of specimen 1. The tuning can be realized by applying an appropriate magnetic field to target channels or to all channels simultaneously.

## 4. Concluding Remarks

In this study, a new flexible beam structure with a tunable stiffness was fabricated by 3D printing. Its field-dependent stiffness was experimentally validated. To tune the stiffness, the MRF was embedded in the 12 channels (specimen 1) and three channels (specimen 2) in the skin of the TPU. The stiffness of specimen 1 could be increased by approximately 22.9% and 13.9% by applying a magnetic field to the MRF domain by using PMs attached on the top and bottom for 1 mm and 2 mm strokes, respectively. By changing the position of PMs, the stiffness increment can be altered. Therefore, the optimal locations of PMs need to be determined to maximize the stiffness increment. In the test of specimen 1, the stress–strain results showed that the stress of 110.74 kPa without the magnetic field was increased to 135.82 kPa by PMs attached at both sides of the specimen. On the other hand, in specimen 2, the average stiffness increment ratios from the origin to the strokes of 1 and 2 mm were identified as approximately 39.78% and 23.10% upon the application of the magnetic field, respectively. It is also noted that specimen 2 exhibits considerably a higher increment of stiffness than specimen 1.

These preliminary results presented in this work are self-explanatory in justifying the proof-of-concept of the stiffness change of the proposed soft structure by applying the magnetic field. Further studies are required to develop advanced multifunctional soft structures, based on the use of thinner skins (1–2 mm) with diverse shapes of channels, to provide local and global stiffness tuning and to investigate the haptic function by activating each channel. This study can guide the development of practical products such as stretchable bio-patches which can stimulate specific abnormal nerves and soft robots with wide flexibility ranges.

## Figures and Tables

**Figure 1 micromachines-10-00655-f001:**
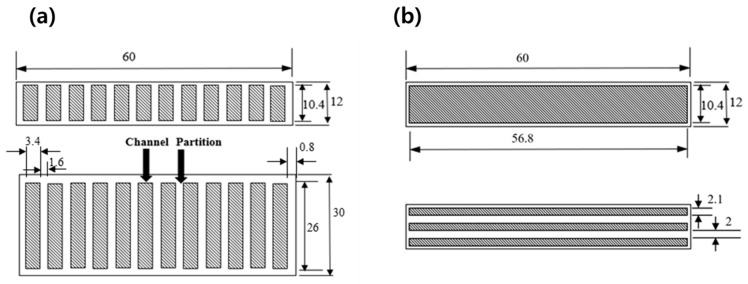
Geometry of the thermoplastic polyurethane (TPU) (unit: mm): (**a**) specimen 1 (12 channels), (**b**) specimen 2 (3 channels).

**Figure 2 micromachines-10-00655-f002:**
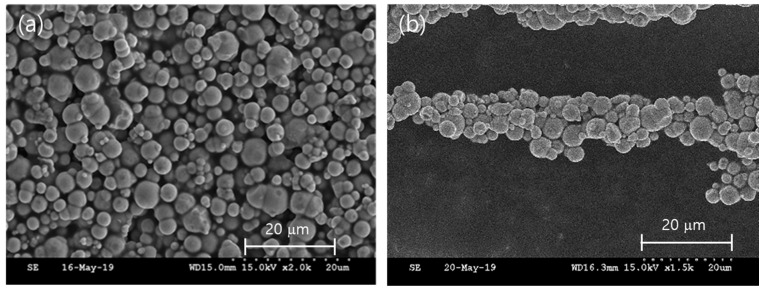
SEM images of the magnetorheological fluid (MRF) (**a**) without the magnetic field and (**b**) under the magnetic field.

**Figure 3 micromachines-10-00655-f003:**
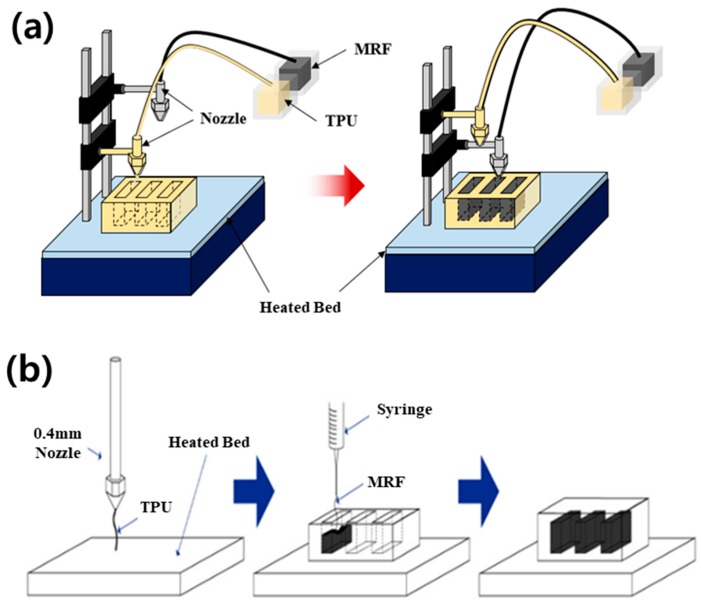
Three-dimensional printing of the soft structure: schematics of the (**a**) equipment and (**b**) manufacturing process.

**Figure 4 micromachines-10-00655-f004:**
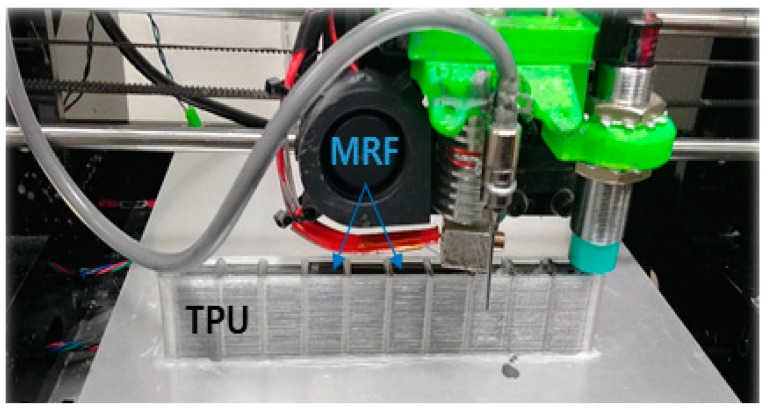
Photograph of the 3D-printing equipment with the MRF and TPU.

**Figure 5 micromachines-10-00655-f005:**
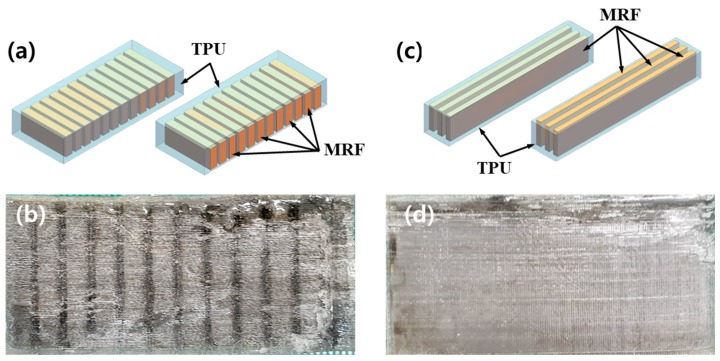
Specimens (TPU and MRF): (**a**,**c**) schematics and (**b**,**d**) surface photographs of specimens 1 and 2, respectively.

**Figure 6 micromachines-10-00655-f006:**
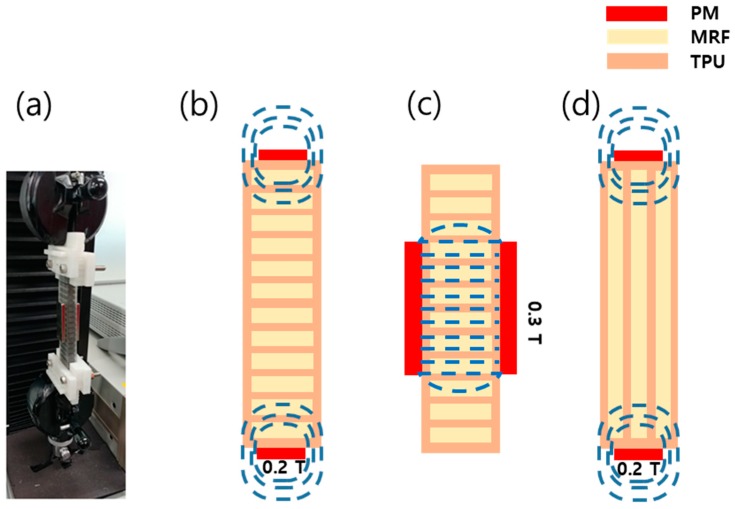
Magnetic field distribution patterns as a function of the permanent magnet (PM) configuration: (**a**) photograph and (**b**) top–bottom, (**c**) lateral (side), and (**d**) top–bottom (with a different cabin structure) configurations.

**Figure 7 micromachines-10-00655-f007:**
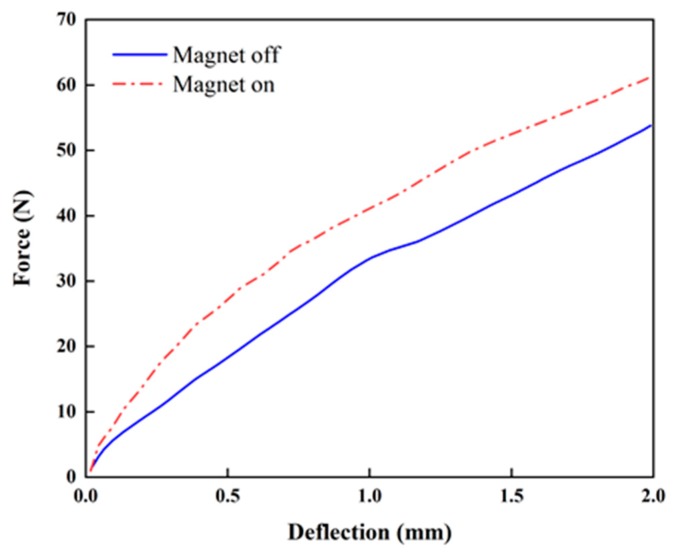
Stiffness variation of specimen 1 (top–bottom PM configuration, Figure 6b).

**Figure 8 micromachines-10-00655-f008:**
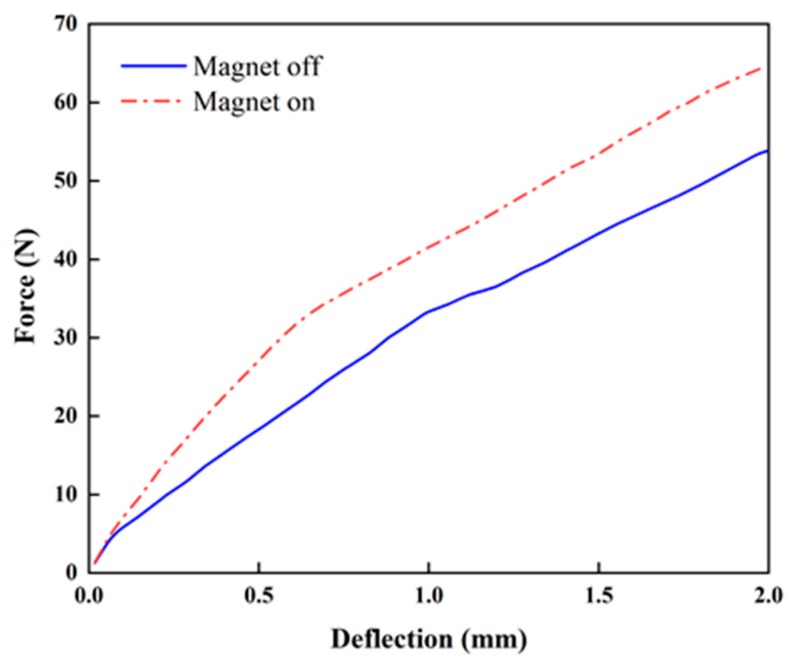
Stiffness variation of specimen 1 (lateral PM configuration, Figure 6c).

**Figure 9 micromachines-10-00655-f009:**
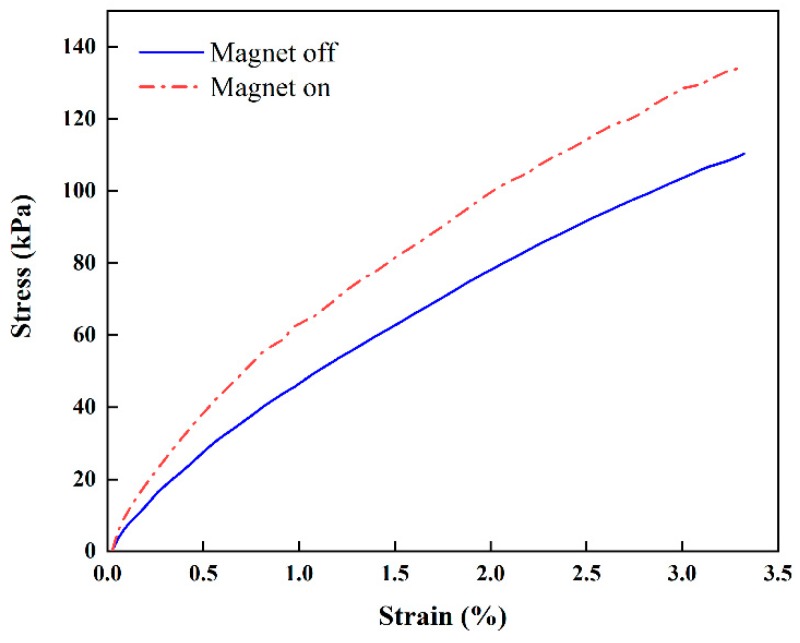
Stress-strain curve of specimen 1 (lateral PM configuration, Figure 6c).

**Figure 10 micromachines-10-00655-f010:**
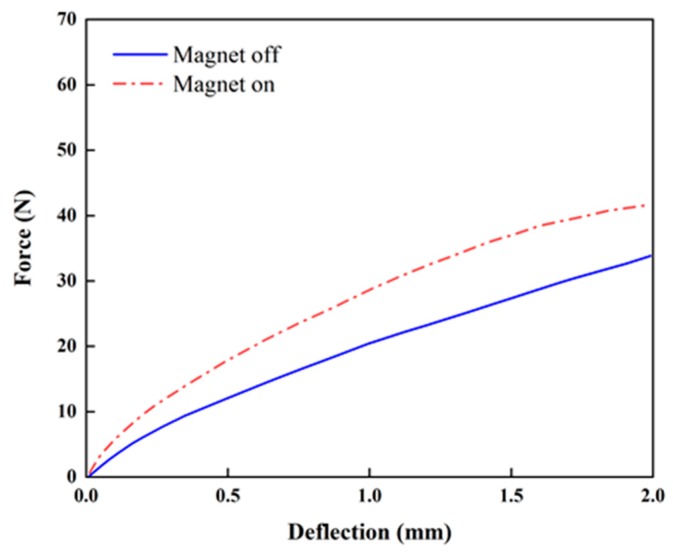
Stiffness variation of specimen 2 (top–bottom PM configuration, Figure 6d).

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
