# Peer review of "3D-Printed Soft Structure of Polyurethane and Magnetorheological Fluid: A Proof-of-Concept Investigation of its Stiffness Tunability"

_micromachines, 2019, doi:10.3390/mi10100655_

Round 1

Reviewer 1 Report

Please check the English language with an language specialist, as the paper contains many spelling and grammatical errors and in some cases phrases have no real meaning On page 3 and 4 you are referring to “heating bed” (including in Fig. 3) – in FDM it is called “heated bed”, therefore please change the term Section 2 - line 83-84 – you refer to some FEM – please explain the process, as this is an important aspect of the experiment design Please explain clearly Figure 2, including the acronym SEM The parameters of the FDM printing process are not given in the paper; as these are rather important can influence considerably the rest of the experiments, the authors should give more details about them: printing speed, layer height, printing temperature, temperature of the heated bed, etc. The authors should consider more specimens, as the validate the conclusion

Reviewer 2 Report

The manuscript titled ‘A 3D printed new soft structure using a polyurethane and a magnetorheological fluid: A proof-of-concept investigation on the stiffness tunability’ seems to be interesting and may get a wide range of readers. But, the manuscript need thorough English correction and clarity in the experiment and discussion.  

Manuscript needs a thorough English and format correction. There are English grammatical and typographical errors in the manuscript, which needs to be corrected. For eg., in line 20, its not ‘prosed soft structure’ it is ‘proposed soft structure’. On line 30, it is not ‘electrical filed’ but ‘electric field’. This is repeated many other places in the manuscript. These are some eg for the typo errors, but there are many in the manuscript. There is a heading for Keywords under the abstract, but could not find any keywords. Figure 2 is the SEM images of MRF, Can you explain how did you do the sample preparation for SEM? How did the authors manage to take SEM images with and without magnetic field inside SEM? Is there any particular reason for using slightly different PM at the top/bottom and sides? What will happen if the PMs are interchanged? According to Figure 6, the magnetic field exert on the specimen is different at different points. How does this affect the stiffness? ‘It is clearly seen that the stiffness is not linear regardless of the magnetic field’. But, from Figures 7 and 8 it looks like the stiffness increases with deflection. Will it get saturate at some point? If so, what is that breaking point? Instead of ‘ratio’, authors written ‘ration’ in many places. It is not clear for me from the manuscript that the stiffness is studied for the entire specimen or at some particular points of the specimen. Please make clarity. ‘The trend is very similar to the result in Figure 7, but the maximum ration of the stiffness increment is higher than that of the case with PM attached at the top and bottom’. How much is the stiffness increment for PM attached at the sides for 1 mm and 2 mm? How much is the error in the measurement? In Figures 7 and 8, On magnet and Off magnet is represented by dotted and continuous lines. But in Figure 9, this order is reversed. It will be nice if authors keep the consistency. Keep consistency in the reference format.

Round 2

Reviewer 1 Report

all my querries were answered

Author Response

All my queries were answered

2.(x) English language and style are fine/minor spell check required 

Answers:

Thank you very much for your positive comments English has been carefully checked one more time in terms of typos, spelling and grammar.

Reviewer 2 Report

‘Electrorheological fluids (ERFs), magnetorheological fluids (MRFs), and magnetorheological elastomers (MREs) are one of the smart materials frequently used for vibration control of flexible structures’. It is not ‘one’ of the smart materials, instead ‘some’ of the smart materials. Authors have introduced a second specimen in the revised manuscript and showed one result of it. I didn’t understand the purpose of this specimen. The second specimen is studied only with PM on top and bottom. Why not on sides? There was stress –strain study and a graph in the old version, which is removed from the revised manuscript. Some of the results and values are changed in the revised manuscript, but these are not corrected in the concluding remarks. Why is the values changed in the revised manuscript?
